# Extending Age Ranges in Breast Cancer Screening in Four European Countries: Model Estimations of Harm-to-Benefit Ratios

**DOI:** 10.3390/cancers13133360

**Published:** 2021-07-04

**Authors:** Nadine Zielonke, Amarens Geuzinge, Eveline A. M. Heijnsdijk, Sirpa Heinävaara, Carlo Senore, Katja Jarm, Harry J. de Koning, Nicolien T. van Ravesteyn

**Affiliations:** 1Department of Public Health, Erasmus MC, University Medical Center Rotterdam, 3015 GD Rotterdam, The Netherlands; h.geuzinge@erasmusmc.nl (A.G.); e.heijnsdijk@erasmusmc.nl (E.A.M.H.); h.dekoning@erasmusmc.nl (H.J.d.K.); n.vanravesteyn@erasmusmc.nl (N.T.v.R.); 2Finnish Cancer Registry, Mass Screening Registry, 00130 Helsinki, Finland; Sirpa.Heinavaara@cancer.fi; 3Epidemiology and screening Unit—CPO, University Hospital Città della Salute e della Scienza, 10126 Turin, Italy; carlo.senore@cpo.it; 4Epidemiology and Cancer Registry, Institute of Oncology, 1000 Ljubljana, Slovenia; kjarm@onko-i.si

**Keywords:** breast cancer screening, harm-to-benefit ratios, microsimulation, overdiagnosis, breast cancer deaths averted, false-positive results

## Abstract

**Simple Summary:**

Breast cancer screening causes harms and benefits. The balance between the two varies by age. By applying microsimulation modelling, we compared several age ranges of screening in four European countries (the Netherlands, Finland, Italy and Slovenia) and evaluated the respective harm-to-benefit ratios. In all countries, adding screening between the ages 45 and 49 or 70 and 74 resulted in more life-years gained and more breast cancer deaths averted, but at the expense of increases in harms. Adapting the age range of breast cancer screening is an option to improve harm-to-benefit ratios in all four countries. The prioritization of considered harms and benefits affects the interpretation of results.

**Abstract:**

The main benefit of breast cancer (BC) screening is a reduction in mortality from BC. However, screening also causes harms such as overdiagnosis and false-positive results. The balance between benefits and harms varies by age. This study aims to assess how harm-to-benefit ratios of BC screening vary by age in the Netherlands, Finland, Italy and Slovenia. Using microsimulation models, we simulated biennial screening with 100% attendance at varying ages for cohorts of women followed over a lifetime. The number of overdiagnoses, false-positive diagnoses, BC deaths averted and life-years gained (LYG) were calculated per 1000 women. We compared four strategies (50–69, 45–69, 45–74 and 50–74) by calculating four harm-to-benefit ratios, respectively. Compared to the reference strategy 50–69, screening women at 45–74 or 50–74 years would be less beneficial in any of the four countries than screening women at 45–69, which would result in relatively fewer overdiagnoses per death averted or LYG. At the same time, false-positive results per death averted would increase substantially. Adapting the age range of BC screening is an option to improve harm-to-benefit ratios in all four countries. Prioritization of considered harms and benefits affects the interpretation of results.

## 1. Introduction

The main benefit of breast cancer screening is a reduction in breast cancer mortality through early detection [1,2,3,4,5,6]. However, screening also causes harm. Important harms associated with breast cancer screening are overdiagnosis and false-positive results [5]. Based on evidence regarding the harms and benefits, the European Commission’s Initiative on Breast Cancer Guidelines Development Group (GDG) strongly recommends inviting women ages 50–69 to mammography screening every two years [7]. Therefore, most European countries adopted biennial screening for breast cancer in this age range [8,9]. Updated evidence on efficacy resulted in extended (conditional) recommendations to triennial or biennial screening for age groups 45–49 and 70–74 in an organized screening programme [7].

Several factors influence the balance between benefits and harms of screening women younger than 50 and older than 69 years. The most important is that breast cancer incidence increases with age [10,11]. Furthermore, the sensitivity of mammography decreases with increasing breast density. Younger women have higher breast density, with lower test sensitivity and more false-positive results [12,13,14]. These two factors might result in smaller benefits and more harms of screening. In contrast, the benefits of screening women ages 70–74 might be limited due to the higher death rate from competing causes with advancing age, thus fewer life-years gained (LYG) and increases in overdiagnosis. 

Unfortunately, there are only a few screening programmes that have accomplished long-term evaluations on the balance between harms and benefits [8]. Often only short-term indicators for benefits and harms are available. Despite several previous studies which assessed the harm-to-benefit-ratios of existing programs for breast cancer [12,15,16], there is no published analysis of the relationship between harms and benefits for varying age ranges and countries.

Therefore, the aim of this study is to assess harm-to-benefit ratios of breast cancer screening vary by age in four European countries. To this end, we calibrated and validated a microsimulation model for each of the four exemplary countries. This study was conducted within the scope of EU-TOPIA. In this project, one exemplary country with high-quality observational data was selected to be representative for each European region (the Netherlands for Western Europe, Finland for Northern Europe, Slovenia for Eastern Europe and Italy for Southern Europe). Using these country-specific models, we estimated the harms and benefits of various screening age ranges. 

## 2. Materials and Methods

### 2.1. Model Overview

The effects of screening for varying age groups were assessed using the Microsimulation Screening Analysis (MISCAN) model [17]. MISCAN simulates individual life histories and assesses the consequences of introducing a screening program on these life histories using the Monte Carlo method. Possible events in the life histories are birth and death of a person, onset of a pre-clinical ductal carcinoma in situ (DCIS), transitions between disease states, participation in screening and screen- or clinical detection of a cancer. (see Appendix A for more information on the MISCAN-Breast structure and underlying assumptions). 

For each of the four countries, we adjusted and calibrated the MISCAN model to reflect differences in population demography (i.e., age distribution of the population and life expectancy), disease risk (i.e., breast cancer incidence and stage distribution) and potential differences in the natural history of breast cancer. In developing each model, we used a specific calibration process (Appendix A). The model optimized a set of unobservable parameters (e.g., stage-specific sensitivity) to match observed data (e.g., detection rates). Thus, we first validated the model versions replicating the data that were used in the calibration process (internal validation). Then, we externally validated the models against best evidence based on a recently published systematic review on breast cancer mortality reductions due to screening [4] (Appendix A). 

### 2.2. Analysis

For each country, we simulated a cohort of 10 million women born in 1975 and followed all women from age 45 until death. First, we simulated the reference screening strategy with biennial screenings from age 50 to 69 years, assuming 100% examination coverage. We assumed 100% to achieve harm and benefit predictions of the tested screening strategies unaffected by external behavioural factors. We then determined the harms and benefits in comparison to no screening. Next, we determined the incremental harms and benefits of extending biennial breast cancer screening to start at age 45 and to stop at age 74. 

### 2.3. Outcomes 

Benefits were expressed as breast cancer deaths averted and LYG. Harms were expressed as false positives and overdiagnoses, calculated as the difference in the number of diagnosed breast cancers in the presence of screening and in the absence of screening, using lifelong follow-up.

For each screening strategy, we determined the following harm-to-benefit ratios by dividing the harms by the benefits: Overdiagnosed breast cancer cases/averted breast cancer deaths;False-positive results/averted breast cancer deaths;Overdiagnosed breast cancer cases/LYG;False-positive results/LYG.

Compared to the reference strategy, an alternative screening strategy could be considered more optimal if one or more harm-to-benefit ratio is smaller.

### 2.4. Sensitivity Analysis

To evaluate how assumptions and parameter values influence the harm-to-benefit ratios and whether the relative differences between strategies change, we performed several sensitivity analyses. First, we assessed the influence of country-specific calibrated values for stage-specific sensitivity by using the highest and the lowest sensitivities and applied them across all countries. Second, we considered the highest and lowest observed referral rates and applied them across all countries. Third, we used observed examination coverage (Table 1) instead of 100%. 

## 3. Results

### 3.1. Model Calibration and Validation

The calibrated models for Slovenia, Finland, the Netherlands and Italy reproduced the country-specific trends in breast cancer incidence and mortality quite well (Appendix A), that is, the simulated model predictions were mostly within the 95% confidence intervals of the corresponding observed outcomes. Subsequently, we validated our model predictions against observed breast cancer mortality reductions due to mammography screening in the Netherlands, Finland and Italy from a systematic review (Appendix A). Due to a lack of studies from Eastern Europe, we validated the Slovenian model by comparing the modelled and observed interval cancer rates (Appendix A).

### 3.2. Outcomes of Different Screening Strategies

If 1000 women underwent biennial mammography between the ages of 50 and 69 (10 screening rounds) and were followed over their lifetimes, the models predicted that around 9000 screening tests would be performed. Compared to a situation without screening, 7 breast cancer deaths would be averted in Slovenia, 8 in Finland, 13 in the Netherlands and 11 in Italy (Table 2). These differences are largely driven by the differences in background incidence rates (Appendix A). The models also predicted that there would be 3 (range 2.5–3.3 across countries) overdiagnosed breast cancer cases per 1000 women when screening between ages 50–69 (Table 2). The overdiagnosed breast cancer cases/breast cancer deaths averted ratio is estimated to range between 0.2 (Italy) and 0.5 (Slovenia). The false-positives/breast cancer deaths averted is estimated to range between 11.6 (the Netherlands) and 45.7 (Italy). Hence, 0.2–0.5 women would be overdiagnosed and 12–46 women would be confronted with a false-positive finding for every woman prevented from dying from breast cancer.

In all countries, adding screening below the age of 50 or after the age of 69 resulted in more life-years gained and more breast cancer deaths averted, but at the expense of increases in harms. For example, screening 1000 women aged 50–74 in Finland is expected to avert 2.4 additional breast cancer deaths, but it would also yield 1.4 additional overdiagnosed cases (Table 2). 

In all countries, the false-positive-related ratios are larger for the younger age ranges and smaller for the older ones compared to reference strategy 50–69. In contrast, the overdiagnosis-related ratios are larger for the older age ranges and tend to be smaller for the strategies where women are screened below the age of 50 (Table 2). 

The percentage change in the harm-to-benefit ratios in comparison to the reference strategy is presented in Figure 1. In all countries, screening women between ages 45–69 would result in smaller overdiagnosis-related ratios. This is particularly pronounced for the ratio of overdiagnosed breast cancer cases to life-years gained. This ratio is 11% (Finland) to 13% (Italy) smaller for the strategy 45–69 than for the reference strategy. On the other hand, the false-positive-related harm-to-benefit ratios for adding screening before the age of 50 or after the age of 69 are less favourable than for screening women between ages 50 and 69.

Of the three alternative strategies, 45–74 is the least optimal age range for screening women in Slovenia, the Netherlands and Italy, as it would lead to an increase in all ratios. In Finland, the least optimal strategy for screening women appears to be 50–75, where the overdiagnosis-related ratios would result in substantial increases (51% and 67%, respectively, Figure 1). 

### 3.3. Sensitivity Analysis

The overdiagnosis-related ratios were relatively insensitive to changing screening test characteristics (Appendix A). However, the false-positive-related ratios were strongly affected by referral rates, leading to an average 14% reduction when applying the lowest age-specific referral rates vs. a two-fold increase when applying the highest age-specific referral rates across all countries. Applying the observed coverage instead of 100% increased the overdiagnosis-related ratios on average by 3% and diminished the false-positive-related ratios by 15%. Varying the values of our input parameters did not affect the magnitude of change of each of the harm-to-benefit ratios when compared to the reference strategy of ages 50–69 (Appendix A).

## 4. Discussion

We were able to calibrate and validate four country-specific microsimulation models in order to investigate long-term outcomes of four breast cancer screening strategies for each European region. Therefore, our results are likely to be relevant to other European countries as well. We found that the ratio of overdiagnosed breast cancer/breast cancer deaths averted could be optimized if screening programs would screen women between ages 45 and 69. By extending the target age range, both the number of life-years gained and breast cancer deaths averted due to screening would increase. However, aside from benefits, extending the screening ages is also associated with additional harms. Of the three alternative strategies, 45–74 is the least optimal age range for screening women in Slovenia, the Netherlands and Italy, while the least optimal range is 50–75 in Finland. 

The impact of the two harms used in our study is considerably different. False-positive results are the most frequent harm of mammography screening, leading to unnecessary testing and an increased benign biopsy rate. In contrast, overdiagnosis is less common, but has a substantial impact. The detection of overdiagnosed cancers turns women into patients, leading to surgery and treatments, which can cause harm and adversely affect quality of life [5]. Moreover, overdiagnosis leads to additional costs and use of healthcare resources. In contrast, false-positive results cause only short-term anxiety, and there is no measurable health utility decrement from this harm [18]. 

It can be debated whether the most serious harm (overdiagnosis) of screening should have equal priority to the most important benefit (the reduction in breast cancer mortality) [19]. However, we believe that the comparability of the two events should be considered. The value of a life saved versus an overdiagnosed case or their consequences are obviously of different magnitude [20]. Being overdiagnosed markedly influences the quality of life of women who experience it as it may cause suffering and anxiety, but it does not affect life expectancy. However, breast cancer screening extends lives [5,21], and therefore many women think overdiagnosis is worth the gain from the potential reduction in breast cancer mortality. In a discrete-choice experiment, Sicsic [22] estimated that women would be willing to accept on average 14.1 overdiagnosed cases and 47.8 false-positive results to avoid one breast-cancer-related death. These results indicate that women consider overdiagnosis 3.4 times as harmful as false-positive results. The ratios we found are well below these thresholds for overdiagnosis per death averted. In all modelled strategies and countries, there are more deaths averted (range 2–3) for every overdiagnosed case. In contrast, two strategies (45–69 and 45–74) in Slovenia and Italia, respectively, have false-positive results per averted breast cancer death above this threshold. 

Our analysis was based on a cohort approach, where women 45 years of age were followed until death. While this approach still considers country-specific all-cause-mortality differences, it eliminates all other external factors such as differences in age structure and makes it possible to solely judge the effect of a change in screening strategy and to compare this effect between countries. However, in reality the differences in age structures between countries might actually play a role and thus affect the decision for a change in screening policy. Of the four countries in this analysis, the Italian population is relatively young, and the Finnish population is relatively old (Appendix A).

To our knowledge, no previous studies analysed the relationship between harms and benefits for varying age ranges and countries. Some studies have specifically assessed the harm-to-benefit ratios for breast cancer screening, but only for the age range 50–69. The EUROSCREEN group estimated 4 overdiagnosed cases and 7 to 9 averted breast cancer deaths per 1000 women, giving a ratio between 0.6 and 0.4 [20,23]. An independent United Kingdom review found an overdiagnosis/breast cancer deaths averted ratio of three to be acceptable [5]. The variation in these results may represent methodological differences, for example in study design and length of follow-up [24]. Our findings for Southern Europe (Italian model) are in line with results of a modelling study for the Basque country, where Arrospide et al. [25] estimated an overdiagnosis/breast cancer deaths averted ratio of 0.3. Van Luijt [26] evaluated the Norwegian Breast Cancer Screening Program in a microsimulation study and estimated a harm-to-benefit ratio of 0.23, whereas we estimated the ratio to be 0.32 for Northern Europe (Finnish model). In a life table model analysis for the United Kingdom, Pashayan [27] assessed that woman who undergo age-based triennial screening between 50 and 69 have twice as many overdiagnosed cases than prevented breast cancer deaths. In contrast, we estimated four times more benefits than harms for Western Europe (Dutch model), despite a shorter screening frequency and higher assumed attendance.

Differences in model estimated ratios likely reflect differences of overdiagnosis estimates, which can vary due to factors such as contrasting definitions of the population at risk. Besides, differences in main model assumptions including the natural history of the disease, differences in length of follow-up and differences in goodness-of-fit of each model can also explain varying estimates [24,28].

Some limitations of this study have to be considered. First, the improvement of prognosis is based on trial data for women age 50–69 years [29,30]. We assumed the same improvement in survival for women outside this age range [31]. Second, our predictions are based on a cohort of women born in 1975. If life expectancy for older women continues to increase in the future, then we might have underestimated the benefits and overestimated the harms of screening for the strategy that screened beyond the age of 69. Third, we maintained the standard two-year screening interval now adopted for the 50 to 69 age range for the alternative strategies, but there is uncertainty about the optimal screening interval for these age ranges, with recommendations ranging between 1 and 3 years. Future work could address different screening intervals by age.

We based our analysis on a comparison to the biennial screening from age 50 to 69 years irrespective of the actual screening policy in each of the four countries. However, the Dutch national breast cancer screening program invites women between 50 and 75 years of age. For the Netherlands, we found that when changing the reference strategy to the current strategy, our findings consistently show that starting screening 5 years earlier would lead to better overdiagnosis-related ratios. This is consistent with a previous microsimulation study based on the same Dutch model showing that digital mammography screening between age 40 and 49 in the Netherlands, in addition to the current screening strategy, is cost-effective [17]. 

The triad of benefits, harms and costs is a key element of health policy decision making. Future research should extend the harm-to-benefit ratios of breast cancer screening to a cost-effectiveness analysis. Such an analysis would consider additional screening effects, such as treatment-related advantages or quality of life, as well as costs. 

## 5. Conclusions

Our study provides insight as to how harm-to-benefit ratios of breast screening programs could be improved by adapting the age range of screened women. Assuming different strategies, this modelling study represents meaningful information on the magnitude of harms and benefits. However, the interpretation of our results depends on how the considered harms and benefits are prioritized by political decision makers.

## Figures and Tables

**Figure 1 cancers-13-03360-f001:**
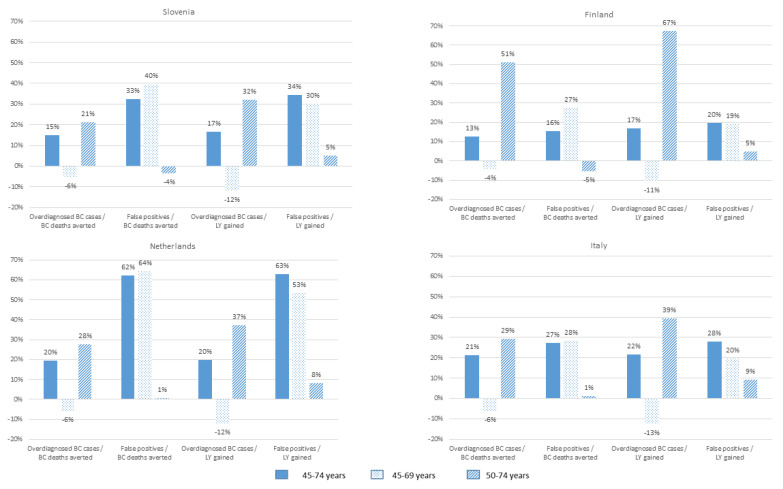
Percentage change in harm-to-benefit ratios in comparison to the reference strategy of ages 50–69, per alternative screening strategy and country. BC: breast cancer; LY: life years.

**Table 1 cancers-13-03360-t001:** Input values for the parametric sensitivity analysis, per country.

Examination Coverage by per Age Group ^1^	Slovenia	Finland	Netherlands	Italy
45–49	54.3% ^2^	85.0% ^2^	75.5% ^2^	59.6% ^2^
50–54	54.3%	85.0%	75.5%	59.6%
55–59	65.0%	85.9%	76.2%	63.2%
60–64	52.4%	86.8%	76.3%	63.9%
65–69	48.8%	73.0%	75.7%	61.5%
70–74	48.8% ^2^	73.0% ^2^	70.1%	61.5% ^2^
Stage-specific sensitivity of digital mammography DCIS	0.726	0.596 ^3^	0.865 ^4^	0.821
Stage-specific sensitivity of digital mammography T1a	0.785	0.811	0.553 ^3^	1 ^4^
Stage-specific sensitivity of digital mammography T1b	0.656	0.761 ^4^	0.481 ^3^	0.717
Stage-specific sensitivity of digital mammography T1c	0.780 ^3^	0.946 ^4^	0.857	0.814
Stage-specific sensitivity of digital mammography T2+	1	1	1	1
Referral rate by age ^5^				
<50	0.040	0.030	0.030 ^3^	0.065 ^4^
>50	0.034	0.028	0.023 ^3^	0.058 ^4^

^1^ The examination coverage of (organised) screening is specified as the proportion (%) of the target population per age group screened in the chosen report year after invitation. These observed parameters stem from the following years: Finland, 2014; Netherlands and Italy, 2015; Slovenia, 2016. ^2^ For those countries that screen women within the age range 50–69, we assumed the same examination coverage for the age groups 45–49 and 70–74 as the nearest age group for which we had observed data. ^3^ This country has the lowest calibrated sensitivity/observed referral rate for the respective cancer stages. ^4^ This country has the highest calibrated sensitivity/observed referral rate for the respective cancer stage. ^5^ The referral rate represents the percentage of participants with abnormal screening results who are referred for further diagnostic testing. This rate depends on the screening protocol adopted for referring women to assessment (i.e., positivity criteria, double vs. single reading), previous opportunistic screening, as well as the quality of screening tests.

**Table 2 cancers-13-03360-t002:** Incremental screening outcomes per country and screening strategy.

Country	Strategy ^1^	Number of Screening Tests	Harms		Benefits		Harm-to-Benefit Ratios			
			Overdiagnosed BC Cases	False Positives	BC Deaths Averted	LY Gained	Overdiagnosed BC Cases/BC Deaths Averted	False Positives/BC Deaths Averted	Overdiagnosed BC Cases/LY Gained	False Positives/LY Gained
Slovenia	50–69 *	9236	3.3	275.8	7.3	96.5	0.5	37.9	0.034	2.9
	45–74	13,723	+1.8	+220.8	+2.6	+32.7	0.5	50.3	0.040	3.8
	45–69	11,696	+0.1	+150.2	+0.8	+18.2	0.4	53.0	0.030	3.7
	50–74	11,264	+1.7	+58.4	+1.9	+14.8	0.5	36.6	0.045	3.0
Finland	50–69 *	9170	2.6	212.3	7.7	105.3	0.3	27.6	0.025	2.0
	45–74	13,632	+1.5	+135.6	+3.2	+38.8	0.4	31.9	0.029	2.4
	45–69	12,034	+0.4	+96.7	+1.4	+24.2	0.3	34.2	0.023	2.4
	50–74	11,183	+1.4	+48.8	+2.4	+19.4	0.4	25.9	0.032	2.1
Netherlands	50–69	8948	3.2	150.1	13.0	185.6	0.2	11.6	0.017	0.8
	45–74	13,288	+1.9	+172.5	+4.2	+59.5	0.3	18.8	0.021	1.3
	45–69	11,388	+0.2	+129.7	+1.8	+40	0.2	19.0	0.015	1.2
	50–74 *	10,848	+1.7	+29.5	+2.5	+19.6	0.3	11.6	0.024	0.9
Italy	50–69 *	9186	2.5	488.5	10.7	152.1	0.2	45.7	0.016	3.2
	45–74	13,657	+1.5	+338.8	+3.5	+49.2	0.3	58.2	0.020	4.1
	45–69	11,641	+0.1	+219.1	+1.4	+32.0	0.2	58.7	0.014	3.8
	50–74	11,203	+1.4	+105.5	+2.1	+17.2	0.3	46.3	0.023	3.5

Model projections for 2020–2075. Screening outcomes are presented per 1000 women aged 45 years and followed over their lifetime. ^1^ Each strategy is compared to no screening. * Current screening strategy. BC: breast cancer; LY: life years. Number of screening rounds per strategy: 50–69: 10; 45–74: 15; 45–69: 12.5; 50–74: 12.5. We assumed 100% adherence to screening strategies including follow-up.

## Data Availability

The data presented in this study are available on request from the corresponding author. The data are not publicly available due to privacy.

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
