# Peer review of "Extending Age Ranges in Breast Cancer Screening in Four European Countries: Model Estimations of Harm-to-Benefit Ratios"

_cancers, 2021, doi:10.3390/cancers13133360_

Round 1

Reviewer 1 Report

This is a very well developed manuscript addressing an important topic, namely the harm/benefit ratio of expanding the age range of mammography screening programs below 50 and above 69. I especially like the description of the methodology, because it is detailed and accessible to the reader who may not be an expert in micro-simulations.

I have only two comments for improvement:

Line 139 the authors mention that the calibrated models "....reproduced trends in breast cancer incidence and mortality quite well (chapter 6, Appendix A)." Can you please be more specific, so that the reader does not have to look up this information in the Appendix.

Line 218 the authors mention that "The value of a life saved versus an overdiagnosed case or their consequences are obviously of different magnitude," Can the authors please elaborate on this statement?

Overall, this reader would have liked  information about sensitivity analyses to be presented in the manuscript and not in the Appendix, such as information in table B1 and Figures B2-B5. Maybe the journal guidelines can accommodate this suggestion.

Author Response

Dear Reviewer 1,

We are grateful to you and the second reviewer for the comments provided for the manuscript "Extending age ranges in breast cancer screening in four European countries: Model estimations of harm-to-benefit-ratios" - Manuscript ID cancers-1216671.

These remarks were very helpful in improving the manuscript and we have revised the manuscript accordingly. Please find attached the revised manuscript.

Changes in the original version of the manuscript are highlighted for added or adjusted sentences in the revised version. There were no additional changes in supplement materials, tables and figures. Please see the replies to the specific comments listed below. 

Best regards,

Nadine Zielonke on behalf of all the co-authors

Reviewer 1, comment 1

Line 139 the authors mention that the calibrated models "....reproduced trends in breast cancer incidence and mortality quite well (chapter 6, Appendix A)." Can you please be more specific, so that the reader does not have to look up this information in the Appendix.

Reply to reviewer comment 1

We thank the reviewer for this critical remark. We developed four different models to reflect differences in population demography (i.e. age distribution of the population and life expectancy), disease risk (i.e. breast cancer incidence and stage distribution) as well as (potential) differences in the natural history between European regions. In calibrating each new country-specific MISCAN-Breast model, we used a specific calibration process composed of four steps as described in Supplements Model description. This process was repeated until the fit with the calibration targets (i.e. country specific breast cancer incidence and mortality, as well as stage distribution) was satisfying, meaning the simulated model predictions were within the 95% confidence intervals of the corresponding observed outcome. That was the case for most age groups in each of the countries.

We revised and clarified as follows: The calibrated models for Slovenia, Finland, the Netherlands and Italy reproduced the country-specific trends in breast cancer incidence and mortality quite well (chapter 6, Appendix A), i.e. the simulated model predictions were mostly within the 95% confidence intervals of the corresponding observed outcome.“(lines 150-153)

Reviewer 1, comment 2

Line 218 the authors mention that "The value of a life saved versus an overdiagnosed case or their consequences are obviously of different magnitude," Can the authors please elaborate on this statement?

Reply to reviewer comment 2

We revised and clarified as follows:

“It can be debated whether the most serious harm (overdiagnosis) of screening should have equal priority to the most important benefit (the reduction in breast cancer mortality) [20]. However, we believe that the comparability of the two events should be considered. The  value of a life saved versus an overdiagnosed case or their consequences are obviously of different magnitude [21]. Being overdiagnosed markedly influences the quality of life of women who experience it as it may cause suffering and anxiety, but the impact on life expectancy is very low, if any. However, breast cancer screening extends lives [5, 22], and therefore many women think overdiagnosis is worth the gain from the potential reduction in breast cancer mortality.” (lines 235-243)

Reviewer 1, comment 3

Overall, this reader would have liked  information about sensitivity analyses to be presented in the manuscript and not in the Appendix, such as information in table B1 and Figures B2-B5. Maybe the journal guidelines can accommodate this suggestion.

Reply to reviewer comment 3

Thank you for this suggestion. We would not object to this change, and asked the editor of the journal whether the journal would find this change feasible.

Reviewer 2 Report

This is a very interesting paper. My few comments are only on the presentation. 

  1. In Simple summary: I think it is misleading to say just under 50 and over 69. Should be specified. 
  2. In Abstract: “Screening women 50-74   … expect Slovenia and Finland” , I find this sentence misleading as Slovenia and Finland clearly will have more over diagnosed BC/ BC death prevented. Conclusion in Abstract a bit weak. My reading of the data is that changing to 45-74 or 50-74 years is not beneficial in any of the countries, and that changing to 45-69 can reduce overdiagnosis but at the price of more false-positive. I think that these are very important messages.
  3. In Discussion: I miss a starting paragraph summarizing he key findings. This should include key findings from each scenario.

Author Response

Dear Reviewer 2

We are grateful to you and the other reviewer for the comments provided for the manuscript "Extending age ranges in breast cancer screening in four European countries: Model estimations of harm-to-benefit-ratios" - Manuscript ID cancers-1216671.

These remarks were very helpful in improving the manuscript and we have revised the manuscript accordingly. Please find attached the revised manuscript.

Changes in the original version of the manuscript are highlighted for added or adjusted sentences in the revised version. There were no additional changes in supplement materials, tables and figures. Please see the replies to the specific comments listed below. 

Best regards,

Nadine Zielonke on behalf of all the co-authors

Reviewer 2, comment 1

In Simple summary: I think it is misleading to say just under 50 and over 69. Should be specified.

Reply to reviewer comment 1

Thank you for pointing out the lack of clarity. We revised and clarified as follows: In all countries, adding screening between the ages 45-49 or 70-74 resulted in more life-years gained and more breast cancer deaths averted, but at the expense of increases in harms.” (lines 17-18)

Reviewer 2, comment 2

In Abstract: “Screening women 50-74   … expect Slovenia and Finland” , I find this sentence misleading as Slovenia and Finland clearly will have more over diagnosed BC/ BC death prevented. Conclusion in Abstract a bit weak. My reading of the data is that changing to 45-74 or 50-74 years is not beneficial in any of the countries, and that changing to 45-69 can reduce overdiagnosis but at the price of more false-positive. I think that these are very important messages.

Reply to reviewer comment 2

According to your suggestion, we revised and clarified the abstract as follows:

“Screening women 45-74 or 50-74 years would be less beneficial in any of the four countries than screening women 45-69, which would result in relatively fewer overdiagnoses per death averted or LYG compared to the reference strategy 50-69. At the same time, false positive results per death averted would increase substantially.” (lines 30-33)

Reviewer 2, comment 3

In Discussion: I miss a starting paragraph summarizing he key findings. This should include key findings from each scenario.

Reply to reviewer comment 3

Thank you for this suggestion.

In this study, we simulated 4 age-specific screening scenarios for 4 countries and estimated 4 hams-to-benefit-ratios. This results in many interesting findings. In order to maintain the readability of the summary, we tried to focus on the most important ones. Therefore, we revised and clarified this section as follows:

“We found that the ratio of overdiagnosed breast cancer /breast cancer deaths averted could be optimized if screening programs would screen women between ages 45-69. By extending the target age-range, both the number of LYG and breast cancer deaths averted due to screening would increase. However, aside from benefits, extending the screening ages is also associated with additional harms. Of the three alternative strategies, 45-74 is the least optimal age-range for screening women in Slovenia, the Netherlands and Italy, while it is 50-75 in Finland.(lines 218-224)